# Identification and Analysis of a CPYC-Type Glutaredoxin Associated with Stress Response in Rubber Trees

**Kun Yuan [1], Xiuli Guo [2], Chengtian Feng [1], Yiyu Hu [1], Jinping Liu [2,*] and Zhenhui Wang [1]**

[1] Key Laboratory of Biology and Genetic Resources of Rubber Tree, Ministry of Agriculture, Rubber Research Institute, Chinese Academy of Tropical Agricultural Sciences, Haikou 571101, China; yuankun628@126.com (K.Y.); fengchengtian@126.com (C.F.); huyy2009@gmail.com (Y.H.); wzh-36@163.com (Z.W.)

[2] Hainan Key Laboratory for Sustainable Utilization of Tropical Bioresources, Tropical Agriculture and Forestry Institute, Hainan University, Haikou 570228, China; aura.guo@vivachek.com

[*] Correspondence: 990785@hainu.edu.cn; Tel.: +86-0898-6696-1263

**Abstract:** Glutaredoxins (GRXs) are a class of small oxidoreductases which modulate various biological processes in plants. Here, we isolated a GRX gene from the rubber tree (*Hevea brasiliensis* Müll. Arg.), named as *HbSRGRX1*, which encoded 107 amino acid residues with a CPYC active site. Phylogenetic analysis displayed that *HbSRGRX1* was more correlated with GRXs from *Manihot esculenta* Crantz. and *Ricinus communis* L. *HbSRGRX1* was localized in the nuclei of tobacco cells, and its transcripts were preferentially expressed in male flowers and in the high-yield variety Reyan 7-33-97 with strong resistance against cold. The expression levels of *HbSRGRX1* significantly decreased in tapping panel dryness (TPD) trees. Furthermore, *HbSRGRX1* was regulated by wounding, hydrogen peroxide ($H_2O_2$), and multiple hormones. Altogether, these results suggest important roles of *HbSRGRX1* in plant development and defense response to TPD and multiple stresses.

**Keywords:** glutaredoxin; subcellular localization; expression; tapping panel dryness; defense response; rubber tree

## 1. Introduction

The rubber tree (*Hevea brasiliensis* Müll. Arg.) from the Euphorbiaceae family, as the major source of natural rubber, is thought to be one of the important industrial trees. The production of natural rubber is facing a serious threat caused by tapping panel dryness (TPD), which is identified by a part or complete cessation of latex flow. It is found that over exploitation, namely over tapping or excessive stimulation by ethephon (ET), might lead to the onset of TPD [1]. Although many researches have tried to reveal the mechanism of TPD, it is still unclear. Numerous genes related to TPD have been identified [2–4]. The onset of TPD is thought to be closely associated with reactive oxygen species (ROS) signaling [4]. The excessive generation of ROS can elicit oxidative damage to lipids, DNA, and proteins, leading to the plant cell death [5]. Plants have evolved various ROS scavenging systems, of which, gutaredoxins (GRXs) are important members that can maintain and modulate cellular redox status with the reducing power of glutathione (GSH) [6]. Increasing evidence indicates that GRXs have antioxidant functions in plant responses to oxidative stress [7–10]. According to the sequences of active sites, plant GRXs contain three main groups, CPY(F)C-, CGFS- and CC-type [6,11]. They take part in the regulation of growth and development, stress responses, and iron-sulfur cluster assembly [9,12–15]. There are 31 GRX members in *Arabidopsis thaliana* (L.) Heynh., 48 in *Oryza sativa*

L., and 36 in *Populus trichocarpa* Torr. & A.Gray ex. Hook. Many researches into CGFS- and CC-type GRX genes have been carried out in plants [8,12,16,17]. However, there are few data about CPYC-type GRXs. Several CPYC-type GRX genes in *Populus trichocarpa* are reported to show differential expression patterns in various organs [11]. Recently, it was found that various treatments could induce the expression of rice *OsGRX20*, a CPYC-type GRX. Further analysis indicates that overexpression of *OsGRX20* in the rice sensitive genotype strikingly increases resistance to bacterial blight [10]. Although many plant GRX genes have been characterized, the biological functions of GRXs in rubber treesremain unknown.

The transcript of the *GRXC9* gene belonging to CC-type class shows decreased expression in TPD trees as compared to healthy ones [4,18], but their biological functions have not been researched in rubber trees. Previously, our proteomic analysis indicated that a GRX protein (ABZ88803.1/ EU295478.1) decreased in the latex of TPD plants compared with healthy ones [19]. Therefore, we postulate that this GRX gene might have critical roles during TPD onset, and it is essential to be further study its function. Here, we sequenced a stress-responsive GRX gene in rubber trees, named as *HbSRGRX1*, and the phylogenetic tree was constructed in the present research. Meanwhile, the expression profiles of *HbSRGRX1* were systematically analyzed in various tissues and varieties, different degrees of TPD trees, wounding, hydrogen peroxide ($H_2O_2$), and various hormone treatments. The results demonstrated that *HbSRGRX1* participated in response to TPD and other various stimuli, suggesting its crucial function in rubber trees.

## 2. Materials and Methods

### 2.1. Plant Materials

The rubber tree clone Reyan 7-33-97 from an experimental field of the Chinese Academy of Tropical Agricultural Sciences in China was used. Seven different tissues from 10-year-old trees were collected to study the tissue-specific expression of *HbSRGRX1*. This type of tree was similarly selected to analyze the influence of $H_2O_2$ and various hormones on *HbSRGRX1* expression. To examine the variety-specific expression of *HbSRGRX1*, 5 different varieties, including Reyan 7-33-97, 7-20-59, 8-79, Reken 523, and PR107 were selected from 10-year-old trees. To study the effect of wounding on *HbSRGRX1* expression, virgin trees from 8-years-old were used. To detect the influence of TPD on *HbSRGRX1*, 24-year-old trees with different TPD degrees (Grade 1, degree <25% tapping panel dry; Grade 2, degree 25% < tapping panel dry < 50%; and Grade 3, degree >50% tapping panel dry) were used. The healthy ones (Grade 0) were used as control (the images of rubber trees from different TPD degrees shown in Figure S1).

### 2.2. Wounding, $H_2O_2$ and Hormones Treatments

For the wounding treatment, eight stainless drawing pins were used to stick into the bark of each tree and left in place as described [20]. Four batches of 10 virgin trees were selected, 3 of which were wounded at 6, 12, 24, and 48 h before the first tapping, and the fourth batch was unwounded as the control. The $H_2O_2$ and hormone treatments were arranged according to the methods of Deng et al. [21] and Long et al. [22], respectively. Three batches (five trees each) were treated with 2% $H_2O_2$, 1.5% ET, 200 μmol/L abscisic acid (ABA), 0.005% methyl jasmonate (MeJA), 200 μmol/L salicylic acid (SA), 66 μmol/L 2,4-dichlorophenoxyacetic acid (2,4-D), 100 μmol/L gibberellic acid (GA$_3$), 200 μmol/L 6-Benzylaminopurine (6-BA) and 100 μmol/L indole-3-acetic acid (IAA). Another one was used as the control. The $H_2O_2$ and hormones were treated at 6, 12, 24, and 48 h.

### 2.3. RNA Isolation and cDNA Synthesis

All latex total RNA was isolated as described [23]. RNA of other tissues was extracted using the RNAprep pure Plant Kit (TIANGEN, Beijing, China). The integrity and concentration of RNA was examined by agarose gel electrophoresis, and a spectrophotometer (Thermo, New York, NY, USA).

The synthesis of First-strand cDNA was conducted with the RevertAid™ First Strand cDNA Synthesis Kit (Fermentas, Waltham, MA, USA)).

### 2.4. ORF Cloning of HbSRGRX1

According to the sequence of EU295478.1, several pairs of primers were designed (Table 1). The open reading frame (ORF) of *HbSRGRX1* was flanked by the pair of primers HbSRGRX1-F (5'-ATGGCGATGACCAAGGCCAAG-3') and HbSRGRX1-R (5'-TTTAAGCAGAAGCCTTAGCAAGAGCTCC-3'). The product was cloned into the pMD18-T vector, and then sequenced.

**Table 1.** Primer sequences.

| Primer Name | Primer Sequence (5'→3') | Use |
|---|---|---|
| HbSRGRX1-F | ATGGCGATGACCAAGGCCAAG | ORF cloning |
| HbSRGRX1-R | TTTAAGCAGAAGCCTTAGCAAGAGCTCC | ORF cloning |
| 1302-HbSRGRX1-F | CTCCCATGGATGGCGATGACCAAGGCCAAG | Subcellular localization analysis |
| 1302-HbSRGRX1-R | CGCACTAGTTTAAGCAGAAGCCTTAGCAAGAGCTCC | Subcellular localization analysis |
| HbSRGRX1-QF | CGTTTCTTCCAATTCTGTTGTCGTT | Real-time PCR analysis |
| HbSRGRX1-QR | CAATGTGCTTGCCACTGATG | Real-time PCR analysis |
| Hb18SrRNA-QF | GCTCGAAGACGATCAGATACC | Real-time PCR analysis |
| Hb18SrRNA-QR | TTCAGCCTTGCGACCATAC | Real-time PCR analysis |

Underline represents the restriction enzyme sites.

### 2.5. Sequence Analyses

The molecular weight (Mw) and isoelectric point (pI) of *HbSRGRX1* were predicted by the ExPASy compute pI/Mw tool. The protein conserved domain was identified with the NCBI (National Center for Biotechnology Information) CDD (Conserved Domain Database) and SMART (Simple Modular Architecture Research Tool) [24]. The protein sequence was aligned by DNAMAN 6. The phylogenetic tree was constructed by MEGA (Molecular Evolutionary Genetics Analysis) 6.06 with neighbor-joining method [25]. A bootstrap test was performed using 1000 replicates.

### 2.6. Subcellular Localization

HbSRGRX1 was localized as described [21]. A pair of primers 1302-HbSRGRX1-F (5'-CTCCCATGGATGGCGATGACCAAGGCCAAG-3') and 1302-HbSRGRX1-R (5'-CGCACTAGTTTAAGCAGAAGCCTTAGCAAGAGCTCC-3') was used to amplify the coding sequence of *HbSRGRX1* (Table 1). The product was cloned into the pMD18-T vector, the correct plasmids and pCAMBIA1302-GFP vector were digested with Nco I and Spe I, respectively. The products were ligated with $T_4$-DNA ligase to obtain the recombinant vector pCAMBIA1302-HbSRGRX1-GFP, which was then transformed into *A. tumefaciens* strain EHA105. The *A. tumefaciens* harboring the recombinant vector was infiltrated into the *N. benthamiana* leaves. Fluorescence signals were examined using a confocal microscope (Fluo View™ FV1000).

### 2.7. Real-time Quantitative PCR (qPCR)

The expression patterns of *HbSRGRX1* were examined by qPCR using the primers HbSRGRX1-QF and HbSRGRX1-QR (Table 1). The 18S rRNA gene (primers: Hb18SrRNA-QF and Hb18SrRNA-QR; Table 1) was used as the internal control. The qPCR was carried out on a LightCycler 2.0 system (Roche Diagnostics, Switzerland) using SYBR Premix Ex Taq™ II (Takara, Dalian, China). The PCR procedures and the calculation of the relative abundance of transcripts were performed as described [20]. All qPCR experiments were reproduced in triplicate, and the values were presented as mean ± SD (Standard Deviation). Figures were drawn by OriginPro 9.0 software (OriginLab Corporation, Northampton, MA, USA).

## 3. Results

### 3.1. ORF Cloning, Sequence Alignment and Phylogenetic Analysis of HbSRGRX1

Primers HbSRGRX1-F and HbSRGRX1-R were designed to clone the ORF of *HbSRGRX1* using latex cDNA of the clone Reyan 7-33-97 as the template. *HbSRGRX1* contained a 324-bp ORF encoding 107 amino acid residues. The putative molecular mass was 11.3 kDa, and the pI was 6.71. HbSRGRX1 contained a conserved motif of the thioredoxin_like superfamily, a CPYC active site at 23–26 amino acids at the N-terminus and GSH binding sites, belonging to CPYC-type class (Figure 1A).

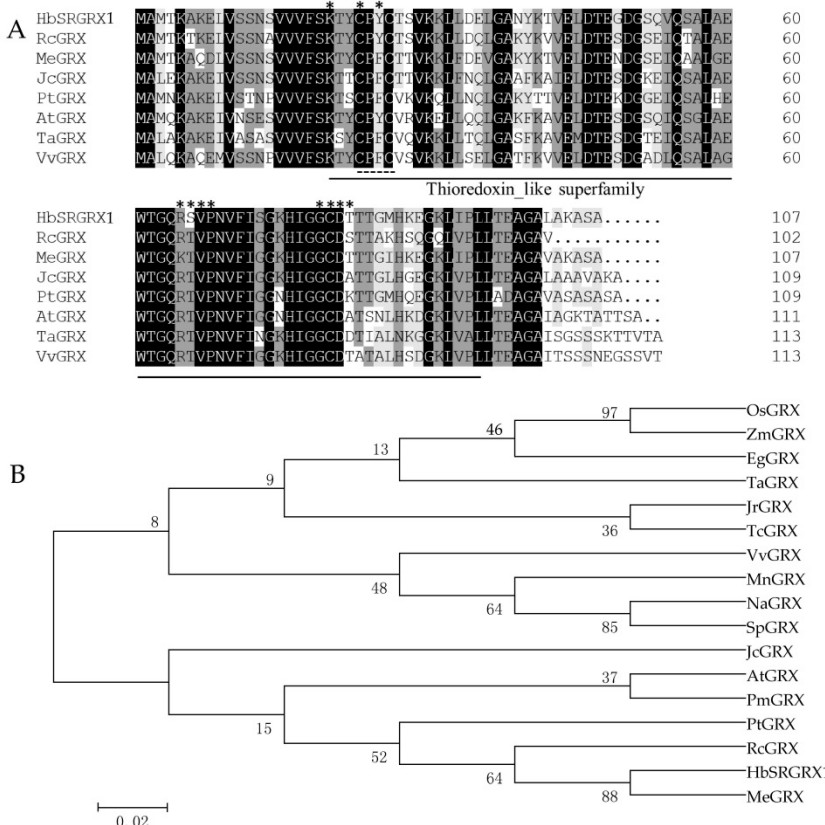

**Figure 1.** Sequence alignment and phylogenetic analyses of *HbSRGRX1* and other plant GRXs. (**A**) Dark and gray indicate identical and similar amino acids, respectively. The straight line, asterisk, and dotted line represent the conserved motif of the thioredoxin_like superfamily, GSH binding site, and active site, respectively. (**B**) The phylogenetic tree was generated by MEGA (Molecular Evolutionary Genetics Analysis) 6.06 using the neighbor-joining method with 1000 bootstrap tests. The scale bar represents the estimated number of amino acid substitutions per site. The accession numbers of GRX proteins are as follows: *Manihot esculenta* (MeGRX, XP 021594601.1), *Ricinus communis* (RcGRX, XP 002524673.1), *Populus trichocarpa* (PtGRX, XP 002298529.2), *Prunus mume* (PmGRX, XP 008230572.1), *Arabidopsis thaliana* (AtGRX, NP 198853.1), *Jatropha curcas* (JcGRX, NP 001295635.1), *Theobroma cacao* (TcGRX, XP 007031532.1), *Juglans regia* (JrGRX, XP 018842397.1), *Solanum pennellii* (SpGRX, XP 015077482.1), *Nicotiana attenuate* (NaGRX, XP 019224739.1), *Morus notabilis* (MnGRX, XP 024029530.1), *Vitis vinifera* (VvGRX, XP 002276266.1), *Triticum aestivum* (TaGRX, AAP80853.1), *Elaeis guineensis* (EgGRX, XP 010940165.1), *Zea mays* (ZmGRX, NP 001158948.1), *Oryza sativa* (OsGRX, XP 015626005.1).

Multiple sequence alignment of *HbSRGRX1* with its related CPY (F) C-type ones from several other plants revealed high identities with the GRXs from *Manihot esculenta* Crantz. *Ricinus communis* L. and *Jatropha curcas* L. (84.11%, 77.57%, and 75.23%, respectively). HbSRGRX1 had 73.39%, 71.17%, 68.42%, and 64.60% identity with GRXs from *Populus trichocarpa*, *Arabidopsis thaliana*, *Vitis vinifera* L.,

and *Triticum aestivum* L., respectively (Figure 1A). To establish the phylogenic relationships among plant GRXs, a phylogenetic tree was constructed between HbSRGRX1 and another 16 plant GRXs in CPY (F) C-type class, using the neighbor-joining method. As shown in Figure 1B, HbSRGRX1 was more closely related to MeGRX (*Manihot esculenta*, XP 021594601.1) and RcGRX (*Ricinus communis*, XP 002524673.1), which belonged to the same family as *Hevea brasiliensis* of Euphorbiaceae. This result demonstrated that the GRXs were highly conserved during evolution.

### 3.2. Subcellular Localization of HbSRGRX1

To further investigate the subcellular localization of *HbSRGRX1* in cells, the *N. benthamiana* leaves were infiltrated with the *A. tumefaciens* harboring pCAMBIA1302-HbSRGRX1-GFP vector and observed with a confocal microscope. The results demonstrated that the fluorescent signal was examined in the nuclei (Figure 2), indicating that HbSRGRX1 was a nucleus located protein.

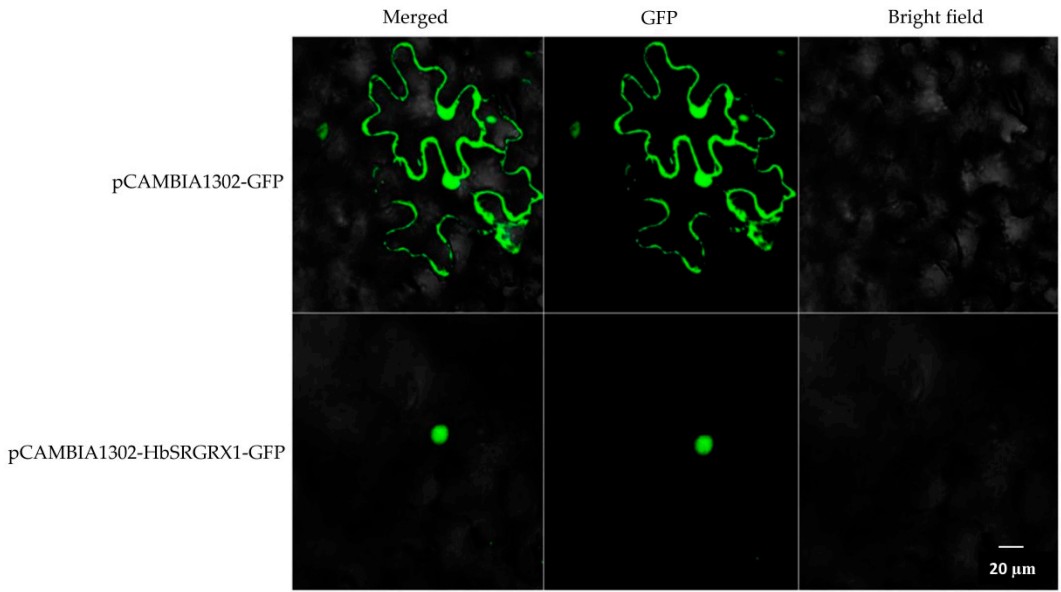

**Figure 2.** Subcellular localization of HbSRGRX1. GFP alone (top row) was localized throughout the whole cell and pCAMBIA1302-HbSRGRX1-GFP (bottom row) in the nuclei of tobacco epidermal cells. Bar = 20 μm.

### 3.3. Expression of HbSRGRX1 in Different Tissues and Varieties

The expression of *HbSRGRX1* was examined in seven tissues, including latex, leaf, bark, male flower, female flower, xylem, and petiole. As shown in Figure 3A, *HbSRGRX1* displayed a tissue-specific expression profile. The expression of *HbSRGRX1* was highest in male flowers, which was strikingly higher than in other tissues ($p < 0.05$), 38.9- and 617.5-fold over its expression in latex and leaf, respectively. In addition, *HbSRGRX1* was differentially expressed in five different varieties of rubber trees, including Reyan 7-33-97, 7-20-59, 8-79, Reken 523, and PR107. The expression level of *HbSRGRX1* was significantly higher in Reyan 7-33-97 than other varieties, the lowest in Reyan 7-20-59 and 8-79 (Figure 3B). Together, the above results indicated that *HbSRGRX1* might have particular functions in different tissues and varieties.

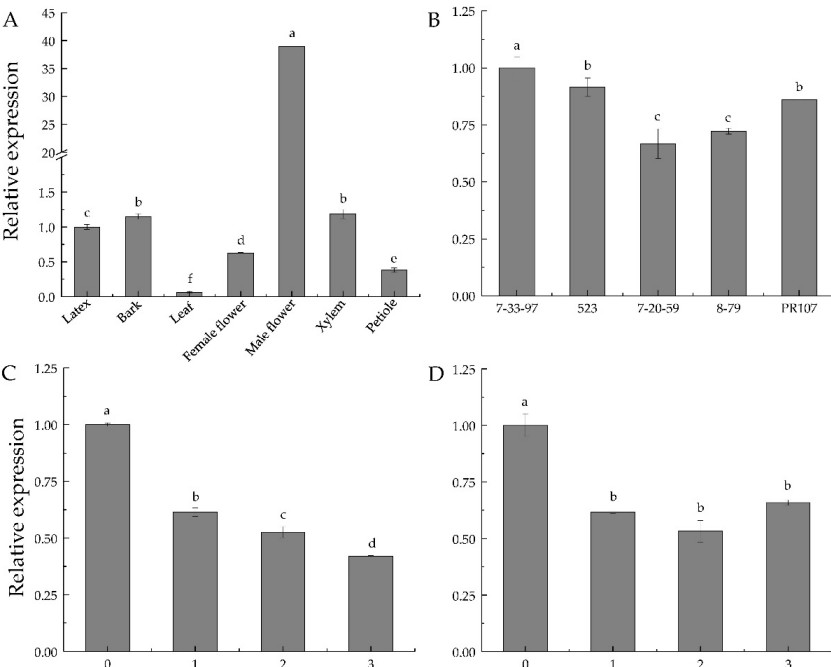

**Figure 3.** Expression profiles of *HbSRGRX1* gene in different tissues and varieties, and in bark and latex from different tapping panel dryness (TPD) levels. Expression profiles of *HbSRGRX1* in different tissues (**A**), barks from different varieties (**B**), bark (**C**), and latex (**D**) from different TPD levels. 0, 1, 2, and 3 indicated healthy trees (Grade 0), Grade 1 (degree <25% tapping panel dry), Grade 2 (degree 25% < tapping panel dry < 50%) and Grade 3 (degree >50% tapping panel dry), respectively. Relative expression was normalized using 18S rRNA gene. Values represent mean ± SD of three independent replicates. Different letters indicate significant difference ($p < 0.05$).

### 3.4. Expression of HbSRGRX1 in Different Degrees of TPD Trees

The TPD of rubber tree is a complicated physiological disorder caused by over tapping or excessive stimulation by ET, seriously affecting natural rubber production [26]. To further analyze the expression of *HbSRGRX1* at the transcriptional level, the bark and latex from different degrees of TPD trees (Grade 0, healthy trees; Grade 1, degree <25% tapping panel dry; Grade 2, degree 25% < tapping panel dry < 50%; and Grade 3, degree >50% tapping panel dry) were collected, respectively. In general, the expression of *HbSRGRX1* was markedly higher in healthy plants than TPD ones (Figure 3C,D). Remarkably, the expression of *HbSRGRX1* in bark showed a trend of constantly significant decline with the degree of TPD rising (Figure 3C), while did not change in latex of different degrees of TPD (1–3) except the healthy control (0) (Figure 3D). These results indicated that the expression levels of *HbSRGRX1* were closely correlated with TPD severities.

### 3.5. Expression of HbSRGRX1 in Response to Different Treatments

Tapping is a kind of mechanical wounding, over tapping or excessive stimulation by ET can elicit oxidative stresses of laticifer cells and the balance between ROS production and scavenging is broken, resulting in ROS burst and the onset of TPD in *Hevea brasiliensis* [1]. Multiple hormones are reported to regulate the expression of GRX genes in rice [10,27]. Thus, the expression profiles of *HbSRGRX1* in latex were systematically investigated in response to wounding, $H_2O_2$, ET, MeJA, ABA, SA, $GA_3$, 2, 4-D, 6-BA, and IAA using qPCR method. As shown in Figure 4, all the treatments regulated the expression of *HbSRGRX1*, but the expression profiles showed obvious variations. After wounding and $H_2O_2$ treatments, *HbSRGRX1* expression indicated an irregular fluctuation and was generally suppressed, with the lowest expression at 6 h and 24 h, respectively (Figure 4A,B). With ET treatment, the expression of *HbSRGRX1* significantly decreased to the lowest level at 24 h (Figure 4C).

Similarly, its expression was rapidly inhibited at 6 h after MeJA treatment and kept a similar expression level from 6 h to 48 h (Figure 4D). *HbSRGRX1* showed similar expression profiles and was quickly induced, all rising to the highest point at 6 h, and then markedly declining at 12 h after ABA, SA, and GA₃ treatments (Figure 4E–G). As for 2, 4-D, 6-BA, and IAA treatments, *HbSRGRX1* exhibited a similar expression trend of first rising and then declining, reaching the peaks at 6 h, 12 h and 24 h, respectively (Figure 4H–J). Taken together, these results demonstrated that *HbSRGRX1* might have distinct functions in response to wounding, $H_2O_2$, and hormones.

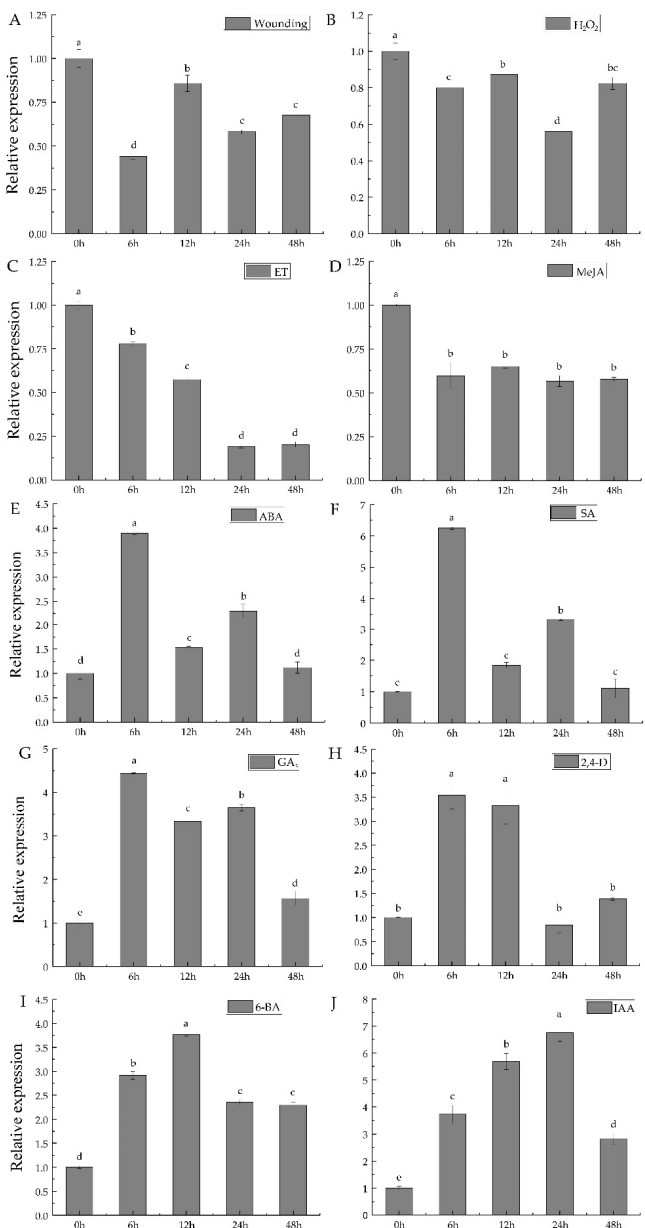

**Figure 4.** Expression profiles of the *HbSRGRX1* gene under different treatments. Expression profiles of *HbSRGRX1* in latex at 6, 12, 24, and 48h after wounding (**A**), hydrogen peroxide ($H_2O_2$) (**B**), ethephon (ET) (**C**), methyl jasmonate (MeJA) (**D**), abscisic acid (ABA) (**E**), salicylic acid (SA) (**F**), gibberellic acid (GA₃) (**G**), 2,4-dichlorophenoxyacetic acid (2,4-D) (**H**), 6-Benzylaminopurine (6-BA) (**I**), and indole-3-acetic acid (IAA) (**J**) treatments. Untreated trees were used as the control (0 h). Relative expression was normalized using 18S rRNA gene. Values represent mean $\pm$ SD of three independent replicates. Different letters indicate significant difference ($p < 0.05$).

## 4. Discussion

Glutaredoxins (GRXs) are a class of small oxidoreductases whose size is commonly 10 to 15 kDa. Many studies have suggested that plant GRXs perform various roles, such as modulating organ development, response to oxidative stress, and hormone signaling [7,12,27]. However, there are few data about the rubber tree GRXs, and their functions are still unclear. In this study, a CPYC-type GRX gene *HbSRGRX1*, encoding 107 amino acids, was isolated from rubber trees. Sequence alignment and phylogenetic analysis indicated that *HbSRGRX1* had high homology with MeGRX and RcGRX of Euphorbiaceae family, suggesting its conservation in the evolution. This research demonstrated that *HbSRGRX1* was localized in the nuclei of tobacco cells. Nuclear localization of GRXs were previously found in other plants [10,27–29]. It is indicated that nuclear localization of a GRX protein belonging to CC-type class from *Arabidopsis thaliana*, ROXY1, is pivotal in controlling petal development [29]. The rice *OsGRX8*, localized in the nuclei and cytosol, is found to take part in response to osmotic, salinity and oxidative stresses [27].

In the present study, the qPCR method was used to systematically investigate the expression profiles of the rubber tree *HbSRGRX1* gene in various tissues and varieties, different degrees of TPD trees, and under multiple treatments. It is reported that the GRXs genes are differentially expressed in various tissues in rice and poplar [10,11,30]. The rubber tree *HbSRGRX1* gene had a tissue-specific expression profile, with the strongest expression in male flowers and weakest in leaves (Figure 3A), suggesting its vital function in the development of male flowers. The CYPC-type *PtrcGrxC3* from *Populus trichocarpa* shows the highest expression in flowers and the lowest in leaves, which is similar to our results [11]. Additionally, the *HbSRGRX1* gene had differential expression in five various varieties, with the highest level in Reyan 7-33-97, followed by Reken 523, PR107, Reyan 8-79, and 7-20-59 (Figure 3B). In the five varieties, Reyan 7-33-97 is a high-yield variety with strong resistance against cold, PR107 and 7-20-59 against wind, while the resistance of Reken 523 and Reyan 8-79 against cold and wind is the weakest. The differential expression of *HbSRGRX1* in various varieties suggested its specific function in defense response of rubber trees.

ROS signaling is reported to be involved in the process of TPD onset, and some ROS-scavenging genes have been identified [4,31]. GRX, as a scavenger of ROS, is found to show down-regulated expression in TPD trees by using transcriptome analysis [4]. Consistently, the expression of the *HbSRGRX1* gene significantly declined in TPD trees in this study (Figure 3C,D). Furthermore, the *HbSRGRX1* expression was repressed by $H_2O_2$ treatment (Figure 4B). According to these results, we speculated that *HbSRGRX1* might play an antioxidant role in TPD response.

Here, we found that the wounding and diverse hormones treatments could also regulate the expression of the *HbSRGRX1* gene (Figure 4). Hormones are known to modulate plant development and environmental stress response. Many studies have demonstrated that the expression of GRX genes is affected by a number of hormones in plants. The *Arabidopsis* GRX480 (also known as GRXC9) may play a key role in SA/JA cross-talk [13], and its expression can be activated by UVB exposure through an SA-dependent and NPR1-independent pathway [32]. Overexpression of rice CC-type *OsGRX8* reduces the sensitivity to plant hormones, ABA, and IAA [27]. The transcripts of rice GRX genes are reported to participate in response to a variety of hormones, including IAA, SA, JA, ABA, cytokinin, and ethylene derivatives [30]. In addition, the expression of the rice *OsGRX20* gene significantly rises after 2, 4-D, JA, SA, and ABA treatments [10]. ET (an ethylene releaser) has been generally used in stimulating latex regeneration in rubber trees. However, the mechanism of ET regulating latex production remains poorly understood. In our study, *HbSRGRX1* showed a significantly down-regulated expression after ET treatment (Figure 4C), suggesting its important function in the ethylene signaling pathway. In plants, JA is known to be a vital hormone involved in controlling a variety of physiological responses [33–36]. It has been demonstrated that laticifer differentiation can be induced by exogenous JA in *H. brasiliensis*, and the number of laticifers is closely correlated with latex production [37]. JA is also indicated to be a key signal molecule in the modulation of rubber biosynthesis [38]. The present results displayed that JA markedly down-regulated the expression of the *HbSRGRX1* gene (Figure 4D). It was possible

that *HbSRGRX1* played an important role in JA-regulating rubber biosynthesis. Besides ET and JA, *HbSRGRX1* expression was induced by hormones ABA, SA, GA$_3$, 2, 4-D, 6-BA, and IAA, reaching the highest level at different times after treatments (Figure 4E-J). Altogether, our results suggest a critical function of *HbSRGRX1* in plant development and response to TPD, wounding, H$_2$O$_2$, and various hormones, and would make a foundation for further characterizing the function of *HbSRGRX1* in *Hevea brasiliensis*.

## 5. Conclusions

HbSRGRX1 encoded a protein for CPYC-type GRX with CPYC active site. *HbSRGRX1* was significantly down-regulated in TPD trees and might play important roles during the onset of TPD. *HbSRGRX1* was preferentially expressed in male flower and might regulate the development of the male flower. *HbSRGRX1* was involved in defense response to wounding and other various stresses. To better understand the relation between *HbSRGRX1* and TPD, it is necessary to study the function of *HbSRGRX1* in-depth in the future.

**Supplementary Materials:** The following are available online at http://www.mdpi.com/1999-4907/10/2/158/s1, Figure S1: Rubber trees from different TPD degrees.

**Author Contributions:** The experiments were designed by Z.W. and J.L. The manuscript was written by K.Y. The experiments were carried out by X.G. The experimental materials were collected by C.F. and Y.H.

**Funding:** This research was funded by the China Agriculture Research System-Natural Rubber (CARS-34-GW5) and the Ministry of Agriculture of China (1630022018009).

**Conflicts of Interest:** The authors declare no conflict of interest.

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
