# Peer review of "Identification and Analysis of a CPYC-Type Glutaredoxin Associated with Stress Response in Rubber Trees"

_forests, doi:10.3390/f10020158_

Round 1

Reviewer 1 Report

The manuscript (Molecular cloning, subcellular localization and expression analysis of a CPYC-type glutaredoxin associated with stress response in rubber tree) by Zhenhui Wang and co-workers provide insights into a glutaredoxin gene isolated from a rubber tree from the Euphorbiaceae family. ORF of HbSRGRX encodes a glutaredoxin protein, which belong to Class I family of glutaredoxins, with CPYC active site. Moreover, subcellular localization and expression was investigated.

The manuscript is interesting and important because it deals with a gene from a rubber tree that is the major commercial source of natural rubber latex. Therefore, molecular characterization of a glutaredoxin gene, which is important in plant development and tolerance against stress, would help lay down the bases for functional characterization and engineering trees more tolerant to stress.

I suggest the following points to improve the MS:

1-in the introduction line 28 and line 29 are irrelevant in the context and to be removed.

2-its better to replace the information about Arabidopsis glutaredoxin by for e.g. Poplar tree glutaredixins since you investigated a tree plant. Moreover, the information provided about Arabidospsis GRXs (nuclear localization) in the discussion are good and shows the link to your results.

3-The quality of figures 3 and 4 needs to be improved.

I recommend accepting the manuscript afters these minor revisions are performed.

Author Response

Dear reviewer,

Thank you very much for your attention and the evaluation and comments from reviewers on our paperMolecular cloning, subcellular localization and expression analysis of a CPYC-type glutaredoxin associated with stress response in rubber tree. We have revised the manuscript according to the detailed suggestions from reviewers. We sincerely hope this manuscript will be finally acceptable to be published on Forests. Thank you very much for all your help and looking forward to hearing from you soon.

Best regards

Sincerely yours

Please find the following Response to the comments of reviewers:

Response to Reviewer 1:

Thanks for your comments on our paper. We have revised our paper according to your comments:

Comment 1: in the introduction line 28 and line 29 are irrelevant in the context and to be removed.

Response: Thanks for the reviewer’s kind suggestion. We removed the introduction line 28 and line 29.

Comment 2: its better to replace the information about Arabidopsis glutaredoxin by for e.g. Poplar tree glutaredixins since you investigated a tree plant. Moreover, the information provided about Arabidospsis GRXs (nuclear localization) in the discussion are good and shows the link to your results.

Response: Thanks for the reviewer’s good suggestion. We replaced the information about Arabidopsis glutaredoxins with Poplar tree glutaredixins. The revised details can be found in Line 72-75 in the revised manuscript.

Comment 3: The quality of figures 3 and 4 needs to be improved.

Response: Thanks for the reviewer’s suggestion. We improved the quality of figures 3 and 4. The revised details can be found in Page 8 (Figure 3) and Page 10 (Figure 4) in the revised manuscript. The attachment also contained the high resolution pictures of Figure 3 and Figure 4.

In addition, the references have been revised correspondingly. If you have any question about this paper, please don’t hesitate to let me know.

Reviewer 2 Report

The authors have analyzed the function of HbSRGRX in the context of tapping panel dryness. They showed that the HbSRGrx is evolutionary conserved, a property which is already known for the many members of the Grx family. They found that HbSRGRX from rubber tree localizes in the nucleus of Tobacco leaves. Since the Grx is quite conserved it would have been interesting to test the several treatments (H2O2, ET…) and to analyze a change in the sub-cellular localization. The most interesting finding is the transcriptional down regulation of HbSRGrx during TPD. However, it remains open whether this is a specific effect.

I think this study has potential, and might help to get a better understanding of TPD and the possible role of Grxs in this syndrome. 

Title:

I would recommend a shorter title without the methodical information (e.g. without molecular cloning) since an association of the Grx with physiological aspects is already given.

Abstract:

line 14: rubber tree (Hevea brasiliensis)… It would be easier for the reader to add the scientific name of the tree at this point.

line 15: …which had a 324-bp open reading frame (ORF)… I would suggest to remove this part because it should be clear that 107 aa are encoded with a certain amount of DNA bases.

line 18: …  and the Reyan 7-33-97, a high-yield variety 19 with strong resistance… change into: and in the high-yield variety Reyan 7-33-97 with a strong resistance against ???.

line 21: … these results suggested critical … change into: these results suggest important roles

…. would make the groundwork for further… change into: will be the basis for further …

Introduction:

The authors give a brief introduction into the field of redoxins. I wonder why they describe the role of Roxys at different points which belong to CC-Type Grxs. Wouldn´t it be more reasonable to focuse on known functions of CPYC-Grx. For the reader it would be important to understand why they explain especially Roxys out of the huge Grx-family and not the more homologues Grx. As a general remark I would encourage the authors to start the introduction with the paragraph starting in line 53, that helps the reader to understand directly why the authors analyzed particularly this HbSRGrx.

I´m not sure whether the authors have information about Isoforms in rubber trees. I guess that there are several isoforms therefore I have the feeling that a number should be added in the name.

What do the authors mean with cloned? Wouldn’t it be more precise to say sequenced ?

line 31: …plants have evolved a various ROS scavenging system… change into: plants have evolved various ROS scavenging systems,

line 31: …plant GRXs contained three main groups… change into: plant Grxs contain three main groups…

line 38: The biological functions of some of them have been characterized [7,12]. Delete this sentence or add more information regarding the different roles.

line 40: …Arabidopsis GRXS13 had…change into: AtGRXS13 has…

Please check carefully the tense! This kind of error appears several times. I will not further correct this kind of error.

line 42: … in transgenic lines knocked down GRXS13…change into: in GRXS13 knock down lines

line 44: ……and their overexpression in transgenic… change into: and the heterologous overexpression in Arabidopsis…

Arabidopsis itself should not be written in italic.

Material and Methods:

The Material and Methods are complete; however, it would be much better for the reader to explain the methods briefly instead of quoting other papers. For example, this is the case in chapter 2.2, in order to understand the research, it would be helpful to add the information how the wounding was applied or how the hormone treatments are realized (how long, how often…).

Line 72:…delete: as the materials.

Results:

I would strongly suggest removing chapter 3.1 and add the essential information into chapter 3.2.

Unfortunately, the resolution of Fig1 is too low. I can’t read the information.

Figure 2 has no legend!! And the signal in brightfield is too low

Figure 3 it would be helpful to add an explanation for the different TPD levels in the panels C and D. Maybe it would be possible to add a few representative pictures in the supplement.

Author Response

Dear reviewer,

Thank you very much for your attention and the evaluation and comments from reviewers on our paperMolecular cloning, subcellular localization and expression analysis of a CPYC-type glutaredoxin associated with stress response in rubber tree. We have revised the manuscript according to the detailed suggestions from reviewers. We sincerely hope this manuscript will be finally acceptable to be published on Forests. Thank you very much for all your help and looking forward to hearing from you soon.

Best regards

Sincerely yours

Please find the following Response to the comments of reviewers:

Response to Reviewer 2:

Thanks for your comments on our paper. We have revised our paper according to your comments:

Comment 1: Title: I would recommend a shorter title without the methodical information (e.g. without molecular cloning) since an association of the Grx with physiological aspects is already given.

Response: Thanks for the reviewer’s good suggestion. The title has been changed into a shorter one “Identification and analysis of a CPYC-type glutaredoxin associated with stress response in rubber tree”.

Comment 2: Abstract: line 14: rubber tree (Hevea brasiliensis)… It would be easier for the reader to add the scientific name of the tree at this point.

Response: Thanks for the reviewer’s kind suggestion. The “rubber tree” in the abstract has been changed into “rubber tree (Hevea brasiliensis)”. The revised details can be found in Line 15 in the revised manuscript.

Comment 3: Abstract: line 15: …which had a 324-bp open reading frame (ORF)… I would suggest to remove this chapter because it should be clear that 107 aa are encoded with a certain amount of DNA bases.

Response: Thanks for the reviewer’s suggestion. We have removed “which had a 324-bp open reading frame (ORF)”. The revised details can be found in Line 16 in the revised manuscript.

Comment 4: Abstract: line 18: and the Reyan 7-33-97, a high-yield variety 19 with strong resistance… change into: and in the high-yield variety Reyan 7-33-97 with a strong resistance against ???.

Response: Thanks for the reviewer’s kind suggestion. We have changed “and the Reyan 7-33-97, a high-yield variety 19 with strong resistance” into “and in the high-yield variety Reyan 7-33-97 with a strong resistance against cold”. The revised details can be found in Line 20-21 in the revised manuscript.

Comment 5: Abstract: line 21: … these results suggested critical … change into: these results suggest important roles. …. would make the groundwork for further… change into: will be the basis for further …

Response: Thanks for the reviewer’s suggestion. We have changed “… these results suggested critical …” into “these results suggest important roles”. We have changed “…. would make the groundwork for further…” into “will be the basis for further …”. The revised details can be found in Line 24-26 in the revised manuscript.

Comment 6: Introduction: The authors give a brief introduction into the field of redoxins. I wonder why they describe the role of Roxys at different points which belong to CC-Type Grxs. Wouldn´t it be more reasonable to focuse on known functions of CPYC-Grx. For the reader it would be important to understand why they explain especially Roxys out of the huge Grx-family and not the more homologues Grx. As a general remark I would encourage the authors to start the introduction with the paragraph starting in line 53, that helps the reader to understand directly why the authors analyzed chaptericularly this HbSRGrx.

Response: Thanks for the reviewer’s good suggestion. The introduction has been started with the paragraph starting in line 53 in original manuscript. Furthermore, we have focused on the known functions of CPYC-Type Grxs, not CC-Type Grxs. This will help the reader to understand directly why we analyzed chaptericularly this HbSRGrx. The revised details can be found in Line 57-79 in the revised manuscript.

Comment 7: Introduction: I´m not sure whether the authors have information about Isoforms in rubber trees. I guess that there are several isoforms therefore I have the feeling that a number should be added in the name.

Response: Thanks for the reviewer’s suggestion. The gene name “HbSRGRX” has been changed into “HbSRGRX1” throughout the text in the revised manuscript.

Comment 8: Introduction: What do the authors mean with cloned? Wouldn’t it be more precise to say sequenced ?

Response: Thanks for the reviewer’s kind suggestion. The “cloned” has been changed into “sequenced” in the revised manuscript. The revised details can be found in Line 86 in the revised manuscript.

Comment 9: Introduction: line 31: …plants have evolved a various ROS scavenging system… change into: plants have evolved various ROS scavenging systems.

Response: Thanks for the reviewer’s kind suggestion. The sentence “…plants have evolved a various ROS scavenging system” has been changed into “plants have evolved various ROS scavenging systems”. The revised details can be found in Line 65 in the revised manuscript.

Comment 10: Introduction: line 31: …plant GRXs contained three main groups… change into: plant Grxs contain three main groups…

Response: Thanks for the reviewer’s kind suggestion. The sentence “plant GRXs contained three main groups” has been changed into “plant Grxs contain three main groups”. The revised details can be found in Line 69-70 in the revised manuscript.

Comment 11: Introduction: line 38: The biological functions of some of them have been characterized [7,12]. Delete this sentence or add more information regarding the different roles.

Response: Thanks for the reviewer’s good suggestion. The sentence “The biological functions of some of them have been characterized [7,12]” has been deleted in the revised manuscript.

Comment 12: Introduction: line 40: …Arabidopsis GRXS13 had…change into: AtGRXS13 has…Please check carefully the tense! This kind of error appears several times. I will not further correct this kind of error.

Response: Thanks for the reviewer’s kind suggestion. We have checked carefully the tense throughout the text and some errors have been found and corrected in the revised manuscript.

Comment 13: Introduction: line 42: …in transgenic lines knocked down GRXS13…change into: in GRXS13 knock down lines

Response: Thanks for the reviewer’s suggestion. According to the advice of the comment 6, this content has been removed.

Comment 14: Introduction: line 42: …in transgenic lines knocked down GRXS13…change into: in GRXS13 knock down lines

Response: Thanks for the reviewer’s suggestion. According to the advice of the comment 6, this content has been removed.

Comment 15: Introduction: line 44: …and their overexpression in transgenic… change into: and the heterologous overexpression in Arabidopsis…

Response: Thanks for the reviewer’s suggestion. According to the advice of the comment 6, this content has been removed.

Comment 16: Material and Methods: The Material and Methods are complete; however, it would be much better for the reader to explain the methods briefly instead of quoting other papers. For example, this is the case in chapter 2.2, in order to understand the research, it would be helpful to add the information how the wounding was applied or how the hormone treatments are realized (how long, how often…).

Response: Thanks for the reviewer’s good suggestion. The methods of the wounding and hormone treatments have been added to the text and the revised details can be found in Line 107-117 in the revised manuscript. The method of phylogenetic analysis has been added to Chapter 2.5 and the revised details can be found in Line 136-137 in the revised manuscript. The qPCR method has been added to Chapter 2.7 and the revised details can be found in Line 153-157 in the revised manuscript.

Comment 17: Material and Methods: Line 72:…delete: as the materials 

Response: Thanks for the reviewer’s suggestion. We deleted “as the materials”. The revised details can be found in Line 96 in the revised manuscript.

Comment 18: Results: I would strongly suggest removing chapter 3.1 and add the essential information into chapter 3.2.

Response: Thanks for the reviewer’s good suggestion. We have removed chapter 3.1 and add the essential information into chapter 3.2. The revised details can be found in Line 159-167 in the revised manuscript.

Comment 19: Results: Unfortunately, the resolution of Fig1 is too low. I can’t read the information. Figure 2 has no legend!! And the signal in brightfield is too low

Response: Thanks for the reviewer’s kind suggestion. We improved the quality of Figures 1 and 2. The revised details can be found in Page 6 (Figure 1) and Page 7 (Figure 2) in the revised manuscript. The attachment also contained the high resolution pictures of Figure 1 and Figure 2. In addition, the legend of Figure 2 has been added to revised manuscript. The revised details can be found in Line 200-202 in the revised manuscript.

Comment 20: Results: Figure 3 it would be helpful to add an explanation for the different TPD levels in the panels C and D. Maybe it would be possible to add a few representative pictures in the supplement.

Response: Thanks for the reviewer’s good suggestion. We have added four representative pictures (Figure S1) indicating the rubber trees from different TPD degrees in the supplementary material. The attachment contains the details of the supplementary material.

In addition, the references have been revised correspondingly. If you have any question about this paper, please don’t hesitate to let me know.

Reviewer 3 Report

The manuscript by Yuan et al., aims at investigating the expression profile of a specific GRX gene from rubber tree following exposure to multiple treatments with emphasis on TPD. In addition, the authors cloned the protein, performed sequence analysis and built a phylogenetic tree aimed at analyzing the conservation of this GRX isoform. Although the manuscript is well presented and contains interesting data, there are some points that to need to be addressed to improve the clarity and to further sustain authors’ conclusions.

The introduction must be improved. Here some points that need to be addressed.

- The concept “To eliminate the oxidative damage” has a poor meaning and should be revised.

- The nomenclature of GRX proteins has been recently described in a review (Zaffagnini et al., ARS 2018; DOI: 10.1089/ars.2018.7617). Please refer to this work to make homogenous GRX nomenclature.

- In order to clarify the description of GRX functions, the authors should clearly state the type to which the different glutaredoxins belong.

- The authors should indicates what GRX genes are modulated in TPD trees. The proteomic study employed by the authors must be cited. The definition “multiple treatments” requires further clarification.

Material and Methods

- Some experimental procedures must be explained without quoting previous papers

Results

- The concept of similarity is misleading and identity would be preferred when comparing protein sequences (Figure 1A). What is the CPY (F) C-type GRX class? This should be mentioned earlier in the text. An experiment with nucleus staining is demanded to clearly establish nuclear localization of HsSRGRX.  

- Based on expression analyses, the authors conclude that HsSRGRX might have particular functions in different tissues and varieties. Although this statement is correct, additional points could be discussed taking into account the possible role of GRX in the different tissues (e.g. almost no expression in leaves), in the different varieties (e.g. what are the differences among the selected varieties?), etc. In addition to methods, the degree of TPD should be mentioned again in the results.

- In reviewer’s opinion, it is not clear why the expression profile of GRX after TPD was not investigated at different time points as done for other treatments. This would have been worth of investigation. In addition, the effects of wounding, H2O2, and hormones on GRX expression is somehow puzzling and the authors should focus on some of them introducing, even briefly, the principles of these analyses.  

Discussion/Conclusions

- The authors should clarify to what GRX types (please consider replacing type by class) ROXY and OsGRX8 belong to. How can the authors support the following statements (P.9, lines 260-261) “It was speculated that …stress response”?

- The conclusion should be revised as it seems a summary of what presented earlier. In this section, the authors should bring the readers to concluding statements along with future perspectives.  

Minor points

- Paragraph 3.1 and 3.2 can be fused to integrate the information in one single paragraph

- Figure quality must be improved. Figure 2 lacks the legend, and Figure 3 and 4 lack information to become self-explanatory (TPD degrees, etc.)

- A careful re-reading of the manuscript is mandatory to fix unclear sentences

Author Response

Dear reviewer,

Thank you very much for your attention and the evaluation and comments from reviewers on our paperMolecular cloning, subcellular localization and expression analysis of a CPYC-type glutaredoxin associated with stress response in rubber tree. We have revised the manuscript according to the detailed suggestions from reviewers. We sincerely hope this manuscript will be finally acceptable to be published on Forests. Thank you very much for all your help and looking forward to hearing from you soon.

Best regards

Sincerely yours

Please find the following Response to the comments of reviewers:

Response to Reviewer 3:

Thanks for your comments on our paper. We have revised our paper according to your comments:

Comment 1: introduction: The concept “To eliminate the oxidative damage” has a poor meaning and should be revised.

Response: Thanks for the reviewer’s kind suggestion. We deleted the sentence “To eliminate the oxidative damage”. The revised details can be found in Line 65 in the revised manuscript.

Comment 2: introduction: The nomenclature of GRX proteins has been recently described in a review (Zaffagnini et al., ARS 2018; DOI: 10.1089/ars.2018.7617). Please refer to this work to make homogenous GRX nomenclature.

Response: Thanks for the reviewer’s kind suggestion. We have referred to this nomenclature of GRX proteins from Zaffagnini et al. Plant GRXs contain three main groups, CPY(F)C-, CGFS- and CC-type. This is added to the introduction. The revised details can be found in Line 69-70 in the revised manuscript.

Comment 3: introduction: In order to clarify the description of GRX functions, the authors should clearly state the type to which the different glutaredoxins belong.

Response: Thanks for the reviewer’s kind suggestion. We have revised the statement about the type to which the different glutaredoxins belong. The revised details can be found in Line 69-76 in the revised manuscript.

Comment 4: introduction: The authors should indicate what GRX genes are modulated in TPD trees. The proteomic study employed by the authors must be cited. The definition “multiple treatments” requires further clarification.

Response: Thanks for the reviewer’s good suggestions. The transcript of GRXC9 gene belonging to CC-type class shows decreased expression in TPD trees as compared to healthy ones. The gene name and the type have been added to the text and the revised details can be found in Line 80 in the revised manuscript. The results from our proteomic study have been cited in the introduction and the revised details can be found in Line 81 and Line 396-398 in the revised manuscript. The definition “multiple treatments” is clarified by “wounding, H2O2 and various hormones” and the revised details can be found in Line 89-90 in the revised manuscript.

Comment 5: Material and Methods: Some experimental procedures must be explained without quoting previous papers.

Response: Thanks for the reviewer’s good suggestions. The methods of the wounding and hormone treatments have been added to the text and the revised details can be found in Line 107-117 in the revised manuscript. The method of phylogenetic analysis has been added to Chapter 2.5 and the revised details can be found in Line 136-137 in the revised manuscript. The qPCR method has been added to Chapter 2.7 and the revised details can be found in Line 153-157 in the revised manuscript.

Comment 6: Results: The concept of similarity is misleading and identity would be preferred when comparing protein sequences (Figure 1A). What is the CPY (F) C-type GRX class? This should be mentioned earlier in the text. An experiment with nucleus staining is demanded to clearly establish nuclear localization of HsSRGRX.  

Response: Thanks for the reviewer’s good suggestions. The “similarity” has been changed into “identity” when comparing protein sequences and the revised details can be found in Line 168 in the revised manuscript. We add the definition “CPY (F) C-type” GRX class in the introduction and the revised details can be found in Line 69 in the revised manuscript. It is important that an experiment with nucleus staining is performed to clearly establish nuclear localization of HsSRGRX. However, we are very sorry that the experiment is very difficultly carried out due to the limitation of the materials.

Comment 7: Results: Based on expression analyses, the authors conclude that HsSRGRX might have chaptericular functions in different tissues and varieties. Although this statement is correct, additional points could be discussed taking into account the possible role of GRX in the different tissues (e.g. almost no expression in leaves), in the different varieties (e.g. what are the differences among the selected varieties?), etc. In addition to methods, the degree of TPD should be mentioned again in the results.

Response: Thanks for the reviewer’s good suggestions. The possible role of HbSRGRX1 in the different tissues has been analyzed in the discussion and the revised details can be found in Line 282-285 in the revised manuscript. In the selected 5 varieties, Reyan 7-33-97 is a high-yield variety with strong resistance against cold, PR107 and 7-20-59 against wind, while the resistance of Reken 523 and Reyan 8-79 against cold and wind is the weakest. The details have been added to the discussion in Line 287-291 in the revised manuscript. The statement about the degree of TPD has been added to the results in Line 218-219 in the revised manuscript.

Comment 8: Results: In reviewer’s opinion, it is not clear why the expression profile of GRX after TPD was not investigated at different time points as done for other treatments. This would have been worth of investigation. In addition, the effects of wounding, H2O2, and hormones on GRX expression is somehow puzzling and the authors should focus on some of them introducing, even briefly, the principles of these analyses.

Response: Thanks for the reviewer’s suggestions. TPD is a complicated physiological disorder caused by over tapping or excessive stimulation by ethephon. The time points of TPD onset for different trees are different, and thus the expression profile of GRX cannot be investigated at different time points as done for other treatments. The principles analyzing the effects of wounding, H2O2, and hormones on GRX expression were stated in Chapter 3.5 in Line 237-240 in the revised manuscript.

Comment 9: Discussion: The authors should clarify to what GRX types (please consider replacing type by class) ROXY and OsGRX8 belong to. How can the authors support the following statements (P.9, lines 260-261) “It was speculated that …stress response”?

Response: Thanks for the reviewer’s kind suggestion. ROXY and OsGRX8 both belong to CC-type class. The details have been added to the Line 273 and Line 305 in the revised manuscript. We have deleted the sentence “It was speculated that …stress response”. The revised details can be found in Line 276-277 in the revised manuscript.

Comment 10: Conclusions: The conclusion should be revised as it seems a summary of what presented earlier. In this section, the authors should bring the readers to concluding statements along with future perspectives.  

Response: Thanks for the reviewer’s good suggestion. We have revised the conclusion in Chapter 5. The revised details can be found in Line 326-331 in the revised manuscript.

Comment 11: Minor points: Paragraph 3.1 and 3.2 can be fused to integrate the information in one single paragraph

Response: Thanks for the reviewer’s suggestions. We have fused chapter 3.1 and chapter 3.2. The revised details can be found in Line 159-167 in the revised manuscript.

Comment 12: Minor points: Figure quality must be improved. Figure 2 lacks the legend, and Figure 3 and 4 lack information to become self-explanatory (TPD degrees, etc.)

Response: Thanks for the reviewer’s suggestions. We have improved the quality of Figures 1, 2, 3 and 4. The revised details can be found in Page 6 (Figure 1), Page 7 (Figure 2), Page 8 (Figure 3) and Page 10 (Figure 4) in the revised manuscript. The attachment also contained the high resolution pictures of Figure 1-4. In addition, the legend of Figure 2 has been added to revised manuscript. The revised details can be found in Line 200-202 in the revised manuscript. We have added information about TPD degrees in Figure 3 and treatment times in Figure 4, respectively. The revised details can be found in Line 231-233 and Line 258-260 in the revised manuscript.

Comment 13: Minor points: A careful re-reading of the manuscript is mandatory to fix unclear sentences

Response: Thanks for the reviewer’s suggestions. We re-read our manuscript and fix unclear sentences.

In addition, the references have been revised correspondingly. If you have any question about this paper, please don’t hesitate to let me know.

Round 2

Reviewer 2 Report

The manuscript has much improved. I just have a major comment left.

1)      The figures still have a low resolution, but this is maybe be due to preview of the manuscript.

Minor comments:

line 26: … and would will make be the groundwork basis for further characterizing the roles of HbSRGRXHbSRGRX1 in rubber tree. I would suggest to delete this part of the sentence.

line 234: …represented mean ± SD of three independent replicates. Different letters indicated

…represent mean ± SD of three independent replicates. Different letters indicate…

This error appears also in other figure legends.

In general, it could be helpful to recruit a native speaker once to eliminate this kind of mistakes.

Author Response

Manuscript number: 438366

MS Type: Article
Title: Molecular cloning, subcellular localization and expression analysis of a    CPYC-type glutaredoxin associated with stress response in rubber tree

Correspondence Author: liu3305602@163.com (J. Liu)

Dear reviewer,

Thank you very much for your comments and suggestions on our paperMolecular cloning, subcellular localization and expression analysis of a CPYC-type glutaredoxin associated with stress response in rubber tree. We have revised the manuscript according to the suggestions. If you have any question about this paper, please don’t hesitate to contact us. Thank you very much for all your help and looking forward to hearing from you soon.

Best regards

Sincerely yours

Please find the following Response to the comments of reviewers:

Response to Reviewer 2:

Thanks for your comments on our paper. We have revised our paper according to your comments:

Comment 1: The figures still have a low resolution, but this is maybe be due to preview of the manuscript.

Response: Thanks for the reviewer’s good suggestion. The resolution of all the figures is 600 dpi. It may be due to the preview of the manuscript if the figures cannot be still seen clearly.

Comment 2: line 26: … and would will make be the groundwork basis for further characterizing the roles of HbSRGRXHbSRGRX1 in rubber tree. I would suggest to delete this part of the sentence.

Response: Thanks for the reviewer’s kind suggestion. The sentence “… and would will make be the groundwork basis for further characterizing the roles of HbSRGRXHbSRGRX1 in rubber tree” in the abstract has been deleted. The revised details can be found in Line 26 in the revised manuscript.

Comment 3: line 234: …represented mean ± SD of three independent replicates. Different letters indicated…, …represent mean ± SD of three independent replicates; Different letters indicate… This error appears also in other figure legends.

Response: Thanks for the reviewer’s kind suggestion. The sentences “represented mean ± SD of three independent replicates. Different letters indicated” have been changed into “represent mean ± SD of three independent replicates. Different letters indicate”. The revised details can be found in Line 233 in the revised manuscript. This error has also been revised in Figure 4. The revised details can be found in Line 260-261 in the revised manuscript.

.

Reviewer 3 Report

The authors nicely replied to all my concerns and I can understand that some points cannot be addressed due to technical limitations. Based on the current version, I consider the present manuscript worthy of publication in the MDPI journal Forests.

MZ

Author Response

Manuscript number: 438366

MS Type: Article
Title: Molecular cloning, subcellular localization and expression analysis of a    CPYC-type glutaredoxin associated with stress response in rubber tree

Correspondence Author: liu3305602@163.com (J. Liu)

Dear reviewer,

Thank you very much for all your help. If you have any question about this paper, please don’t hesitate to contact us.

Best regards

Sincerely yours